# Identification of the Anti-Infective Aborycin Biosynthetic Gene Cluster from Deep-Sea-Derived *Streptomyces* sp. SCSIO ZS0098 Enables Production in a Heterologous Host

**DOI:** 10.3390/md17020127

**Published:** 2019-02-21

**Authors:** Mingwei Shao, Juying Ma, Qinglian Li, Jianhua Ju

**Affiliations:** 1CAS Key Laboratory of Tropical Marine Bio-resources and Ecology, Guangdong Key Laboratory of Marine Materia Medica, RNAM Center for Marine Microbiology, South China Sea Institute of Oceanology, Chinese Academy of Sciences, 164 West Xingang Road, Guangzhou 510301, China; jianting880720@126.com (M.S.); majunying@scsio.ac.cn (J.M.); liql@scsio.ac.cn (Q.L.); 2University of Chinese Academy of Sciences, 19 Yuquan Road, Beijing 110039, China

**Keywords:** *Streptomyces* sp. SCSIO ZS0098, aborycin, heterogenous expression, antibacterial activity

## Abstract

Aborycin is a ribosomally synthesized member of the type I lasso peptide natural products. In the present study, aborycin was isolated and identified from the deep-sea-derived microbe *Streptomyces* sp. SCSIO ZS0098. The aborycin biosynthetic gene cluster (*abo*) was identified on the basis of genome sequence analyses and then heterologously expressed in *Streptomyces coelicolor* M1152 to effectively produce aborycin. Aborycin generated in this fashion exhibited moderate antibacterial activity against 13 *Staphylococcus aureus* strains from various sources with minimum inhibitory concentrations MICs = 8.0~128 µg/mL, against *Enterococcus faecalis* ATCC 29212 with an MIC = 8.0 µg/mL, and against *Bacillus thuringiensis* with MIC = 2.0 µg/mL. Additionally, aborycin displayed potent antibacterial activity (MIC = 0.5 µg/mL) against the poultry pathogen *Enterococcus gallinarum* 5F52C. The reported *abo* cluster clearly has the potential to provide a means of expanding the repertoire of anti-infective type I lasso peptides.

## 1. Introduction

Aborycin is a representative of the type I lasso peptides and was originally isolated from *Streptomyces* sp. 9440 and *S. griseoflavus* Tü 4072 [1,2]. The structure is composed of a 21 amino acid peptide that is cyclized from the side chain of Asp9 to the N-terminus of Cysl. Two disulfide bonds containing linkages Cys1 → Cys13 and Cys7 → Cys19 form a tricyclic structure consisting exclusively of protein amino acids (Figure 1). Aborycin shares some sequence identity with the HIV-1 protease gp41 which belongs to the aspartic proteinase family, thus inhibiting HIV-1 replication [1,3]. Like aborycin, sviceucin and siamycin are representatives of the type I lasso peptide natural products whose biosynthetic gene clusters (BGCs) have been identified [4] (Figure 2). Feng and co-workers [5] identified the BGC of lasso peptide MS-271 (also termed siamycin) from *Streptomyces* sp. M-271 and established using gene-deletion experiments the indispensability of the biosynthetic enzymes MslA, MslB1, -B2, -C and -H for siamycin production. Additionally, Li and co-workers [6] reported the sviceucin BGC and were successful in heterologously expressing the cluster in *Streptomyces coelicolor*. Notably, Li and co-workers have speculated the presence and details of an aborycin cluster on the basis of bioinformatics. However, since the peptide’s discovery, no gene cluster responsible for aborycin biosynthesis has ever been validated. 

In our continuous efforts to identify anti-infective molecules and to elucidate biosynthetic pathways for secondary metabolites from underexploited marine microorganisms [7,8], we discovered South-China-Sea-derived *Streptomyces* sp. SCSIO ZS0098 for which fermentation extracts showed antibacterial activities against a panel of clinically significant pathogenic bacteria. Herein, we present the isolation, structure elucidation, biological activities and biosynthetic gene cluster for the peptidic natural product aborycin.

## 2. Results

### 2.1. Characterization of Aborycin

In continuing previous initiatives to discover secondary metabolites and their BGCs from South-China-Sea-derived *Streptomyces*, we found a metabolite of *Streptomyces* sp. SCSIO ZS0098 with antibacterial activities against clinically important pathogenic bacteria, including methicillin-resistant *Staphylococcus aureus* (MRSA), *Enterococcus faecalis*, *Bacillus thuringiensis* and *Klebsiella pneumoniae*. Subsequently, large scale fermentation and bioassay-guided fractionation enabled isolation of the compound, and high resolution electrospray ionization mass spectroscopy (HR-ESIMS) data ([M + H]^+^ at *m*/*z* = 2162.8541, calcd for C_97_H_132_N_23_O_26_S_4_: 2162.8575) revealed a molecular formula of C_97_H_13__1_N_23_O_26_S_4_ (Appendix A). ^1^H and ^13^C NMR spectra data suggested the compound in question to be either aborycin or siamycin. We then performed partial hydrolyses (2.0 N HCI, 100 °C, 10 h) which yielded a major fraction (relative molecular masses 887); the structural data of these two peptides were further refined by tandem MS (Appendix A) and found to be consistent with previously published data for aborycin [2]. Consequently, the bioactive species from *Streptomyces* sp. SCSIO ZS0098 was confirmed to be aborycin. Further efforts to characterize both the structure and origins of aborycin were informed by the well-known importance of structure-to-BGC correlations both in native producing organisms as well as heterologous expression systems [9]. To further refine and validate our structural characterization of the isolated compound, we performed a rigorous bioinformatics analysis of *Streptomyces* sp. SCSIO ZS0098. 

### 2.2. Bioinformatics Analysis of *Streptomyces* sp. SCSIO ZS0098

Whole-genome shotgun sequencing of *Streptomyces* sp. SCSIO ZS0098 was accomplished using MiSeq and HiSeq2500 Illumina platforms, (Majorbio Bio-pharm Technology Co., Ltd, Shanghai, China) and the potential of this strain to produce assorted secondary metabolites was analyzed using Antibiotics & Secondary Metabolite Analysis SHell (antiSMASH) version 3.0.5 software, The results of these efforts revealed that the strain houses 17 gene clusters devoted to the synthesis of antimicrobial peptides and polyketides. On the basis of HR-ESIMS and NMR data, we hypothesized that the isolated compound may belong to the lasso peptide family. Notably in this case, the lasso peptide gene cluster identified within the SCSIO ZS0098 genome showed significant similarity to established BGCs associated with sviceucin and siamycin (Figure 2).

Analysis of the lasso peptide gene cluster found in *Streptomyces* sp. SCSIO ZS0098 revealed the presence of 20 ORFs spanning from *orf(-3)* to *orf(+3)* and 22 kb of contiguous genomic DNA containing an aborycin family tricyclic lasso peptide (AboA), a macrolactam synthetase (AboC), a precursor peptide recognition element (AboB1), a cysteine protease (AboB2), and disulfide oxidoreductases AboE and AboF (Table 1); all bore similarities to genes identified in the BGCs for sviceucin and siamycin. The amino acid sequence (AboA as indicated in Figure 2) encodes the synthesis of a 42-residue precursor peptide with a leader region at its N-terminus and a core region proximal the C-terminus containing all 21 amino acids (CLGIGSCNDFAGCGYAVVCFW) that constitute the basic peptide scaffold of aborycin [2,3]. Especially relevant is that in *Streptomyces* sp. SCSIO ZS0098, the core structure of aborycin contains Ile4 and Val17 instead of the Val4 and Ile17 that characterizes siamycin (Figure 1B and Appendix A). This recognition further validated our hypothesis that the identified lasso cluster in SCSIO ZS0098 drives aborycin biosynthesis. Inspired by this logic, we sought to heterologously express the aborycin cluster in *Streptomyces coeliecolor* M1152.

### 2.3. Heterologous Expression of the Candidate Aborycin Gene Cluster in *S. coelicolor* M1152

To heterologously express the candidate aborycin BGC we first generated a genomic cosmid library of *Streptomyces* sp. SCSIO ZS0098 using the Supercos I vector (Agilent Technologies, Santa Clara, CA, USA); 2000 clones were picked and placed into 96-well plates. Three primers targeting *orf(-3)*, *aboB1* and *orf(+5)* were then used to screen for cosmids harboring all of the biosynthetic genes. Cosmid 1512H tested positive for all three primers and was consequently selected for end-sequencing. Sequencing results and bioinformatics analyses revealed that cosmid 1512H contained all 20 ORFs spanning from *orf(-3)* to *orf(+5)*, suggesting that cosmid 1512H encoded for all the machinery needed to biosynthesize aborycin (Figure 2). Cosmid 1512H was then modified by replacing the kanamycin resistance gene within the SuperCos 1 vector with a pSET152AB-derived fragment [10] to yield cosmid 1512H-pSET152AB. The resultant cosmid 1512H-pSET152AB was then transferred into *S. coelicolor* M1152, using standard conjugation methods. The same three primers previously used to target *orf(-3)*, *aboB1* and *orf(+3)* were used to screen for “positive” engineered strains. The candidate strains were fermented using two fermentation media including ISP-2. Subsequent butanone extracts of *S. coelicolor* M1152/1512H were prepared and analyzed by HPLC and HPLC-MS. The results of metabolomics assays revealed that *S. coelicolor* M1152/1512H produced significantly higher amounts of putative aborycin relative to the wild-type (WT) *Streptomyces* sp. SCSIO ZS0098 (Figure 3). Analysis of LC-ESIMS data suggested that a molecular formula of C_97_H_13__1_N_23_O_26_S_4_ ([M + H]^+^ at m/z =2162.8580, calcd for C_97_H_132_N_23_O_26_S_4_: 2162.8575) (Appendix A) was diagnostic of aborycin; analyses of the amino acid sequence of the same species validated this signal’s correlation to the structure of aborycin. Thus, both genomics and metabolomics validated the biosynthetic origins and structural features of aborycin while also highlighting the first reported heterologous production of this lasso peptide representative. 

### 2.4. Antibacterial Activity of the Aborycin

Aborycin has been previously reported to inhibit *Bacillus subtilis* ATCC 6633, *Bacillus brevis* ATCC 9999, *Staphylococcus aureus* ETH 2070, *Pseudomonas saccharophila* ATCC 15946, and *Streptomyces viridochromogenes* TU 57 [2]. This inspired us to screen an expanded panel of pathogens for susceptibility to aborycin. A panel of 35 bacterial targets (including 13 Gram negative bacteria and 22 Gram positive bacteria), were subjected to aborycin using established broth microdilution/activity screening methods [11]. These assays revealed aborycin’s potent antibacterial activities against 14 *S. aureus* (ATCC29213 including 11 clinical isolates and 3 poultry pathogenic *S. aureus*) (see Table 2). With the exception of *S. aureus* MRSA GDE4P037P, aborycin proved active against 13 *S. aureus* with MIC values ranging from 8.0~64 µg/mL. Better yet, aborycin displayed good activity against five clinically relevant and drug-resistant *S. aureus* strains (16339, 6917, 16162, 718306, 745524). Aborycin also showed activity (MIC = 64 µg/mL) against the poultry pathogen *S. aureus* (cfr) GDQ6P012P. In addition, aborycin was active against *Enterococcus faecalis* ATCC 29212 with MIC = 8.0 µg/mL, and *B. thuringiensis* with MIC = 2.0 µg/mL; the compound proved especially active against the poultry-borne pathogenic bacterium *Enterococcus gallinarum* 5F52C with MIC = 0.5 µg/mL. Aborycin displayed only weak antibacterial activities against *Acinetobacter baumannii* (ATCC 19606) as well as four other clinical isolates with MICs ranging from 128~512 µg/mL. Six kinds of *Escherichia coli* (including 4 clinical isolates and 2 poultry pathogens) and *Klebsiella pneumoniae* proved effectively impervious to the effects of aborycin.

## 3. Discussion

Lasso peptides are derived from gene-encoded precursor peptides and are post-translationally modified by dedicated enzymes that cleave the amide bond that tethers the leader and core peptide fragments. This amide scission is believed to be critical for maturation and generation of the lasso topology that characterizes this class of peptide natural products [12]. At the same time, the core peptide confers the characteristic structure of those ribosomally synthesized peptides. Predicated largely on the number of disulfide bonds in the primary structures, lasso peptides are classified as belonging in one of three possible classes [4]. Type I lasso peptides are characterized by a cyclized structure—involving linkage of the Asp9 side chain to the N-terminus of Cysl, and two disulfide bonds linking Cys1 to Cys13, and Cys7 to Cys19—to yield a tricyclic structure containing only proteinaceous amino acids [13]. Type Ⅱ lasso peptides contain no disulfide bonds and type Ⅲ lasso peptides contain only one disulfide linkage. The increasing rates of lasso peptide discovery have attracted attention to these compounds as highly promising drug frameworks by which to generate new enzyme inhibitors and receptor antagonists. Identification of the *abo* cluster, in the current context, is significant by virtue of aborycin’s representation of the type I lasso peptide class and its elusiveness since aborycin’s discovery in the early 1990s. 

Recently, Feng and co-workers [5] identified the BGC of lasso peptide siamycin from *Streptomyces* sp. M-271, whereas Li and co-workers reported the gene cluster encoding sviceucin biosynthesis [6]. We report herein the discovery of the aborycin gene cluster from the deep-sea-derived *Streptomyces* sp. SCSIO ZS0098 and its amenability to heterologous expression and aborycin production in *S. coelicolor*. We have also dramatically expanded our knowledge of aborycin’s antimicrobial activities by screening the compound against a panel of *S. aureus* from various sources, *E. faecalis*, *E. gallinarum*, and *B. thuringiensis*. The findings of these efforts provide significant inspiration for future bioengineering efforts aimed at generating more chimeric and/or non-natural amino acid-containing type I lasso peptides.

## 4. Materials and Methods

### 4.1. General Experimental Procedures

HPLC analyses were performed using an Agilent 1260 Infinity equipment with Diode Array Detector (DAD) (Agilent Technologies, Santa Clara, CA, USA) equipped with a Phenomenex Prodigy ODS (2) column (150 × 4.6 mm, 5 μm; USA). HR-ESIMS data were obtained using a MaXis Q-TOF LC-MS spectrometer (Bruker, Billerica, MA, USA). NMR spectra were obtained with a Bruker Avance 500 spectrometer (Bruker, Billerica, MA, USA). The solvent peak signals of CD_3_OD (*δ*_C_ 49.0 and *δ*_H_ 4.87) were used for calibration. 

### 4.2. Nucleotide Sequence Accession Number

Whole Genome Shotgun sequence data have been deposited at DDBJ/ENA/GenBank under the accession number MKCP00000000. This strain is accessible from China Center for Type Culture Collection (CCTCC), under the accession number of CCTCC M 2016360.

### 4.3. Bacterial Strains, Plasmids

The aborycin producer *Streptomyces* sp. SCSIO ZS0098 was isolated from a deep-sea sediment sample at a depth of 3000 m from the South China Sea and identified as *Streptomyces* sp. on the basis of 16S rRNA sequence comparisons with previously reported sequences in the GenBank database. The strains were streaked and grown at 30 °C on plate containing 15 mL ISP-4 medium for sporulation. *S. coeliecolor* M1152 were selected as heterologous expression hosts for *abo* gene cluster and *Escherichia coli* DH5α was selected as the host for cloning purposes. BW25113/pIJ790 was used as the host for Red/ET-mediated recombination and *E. coli* ET12567/pUZ8002 was used as the DNA donor strain in conjugation with *Streptomyces* sp. SCSIO ZS0098. All *E. coli* strains were cultured at 30 °C or 37 °C in Luria-Bertani (LB) medium. Plasmids, strains and primers used in this study are summarized in Appendix A.

### 4.4. Whole Genome Scanning and Bioinformatics Analysis

Strain *Streptomyces* sp. SCSIO ZS0098 (CCTCC M 2016360) was cultured for 2~3 days in TSB media. Genomic DNA was then extracted using phenol–chloroform extraction and ethanol precipitation. The whole-genome shotgun sequencing of *Streptomyces* sp. SCSIO ZS0098 was performed using Illumina MiSeq and HiSeq2500 platforms by constructing two different gDNA libraries (a paired-end library with insert size of 300~500 bp, and one mate-pair library with insert sizes about 3 kb) at Majorbio Bio-Pharm Technology Co., Ltd, Shanghai, China. After quality filtration, 2152433 high-quality paired-end reads (150 bp) were obtained with an average insert length = 400 bp. The high-quality reads were assembled into 90 contigs with a N50 contig length of 979660 bp using SOAP de novo v2.04 (http://soap.genomics.org.cn/). After gap closing by SOPA Gap Closer v1.12, a draft genome with 21 scaffolds was obtained. The genome of *Streptomyces* sp. SCSIO ZS0098 was subsequently annotated using the Prokaryotic Genome Annotation Pipeline (PGAP) on NCBI (http://www.ncbi.nlm.nih.gov/genome/annotation.prok/).

### 4.5. Genomic Library Construction and Screening

High molecular weight genomic DNA from *Streptomyces* sp. SCSIO ZS0098 was isolated and genomic cosmid library was constructed using SuperCos 1 according to established protocols [14]. About 2000 clones were picked and placed into 96-well plates and stored at −80 °C. Then three primers targeting *orf(-3)*, *aboB1* and *orf(+5)* were used for screening the cosmid harboring all candidate gene cluster components.

### 4.6. Construction of the Aborycin Gene Cluster Heterologous Expression Strains

The cosmid 1512H harboring the aborycin gene cluster was first transferred into *E. coli* BW25113/pIJ790. Then an *aac(3)IV-oriT-intφ*C31 cassette was amplified from plasmid pSET152AB after digestion by *Bam*HI/*Eco*RI to replace the kanamycin resistance gene on cosmid 1512H via Red/ET-mediated recombination [14]. The modified cosmid 1512H was transferred into *E. coli* ET12567/pUZ8002 and transferred into *S. coelicolor* M1152, respectively, using conjugation. The same three primers targeting *orf(-3)*, *aboB1* and *orf(+5)* were used for screening the positive engineered strains.

### 4.7. Metabolite Analyses of WT and Recombinant Heterologous abo Cluster Expression Strains

The WT *Streptomyces* sp. SCSIO ZS0098, heterologous expression strains and mutant strains were first grown on ISP-4 medium at 28 °C for 5–7 days to achieve sporulation and then inoculated into a 250 mL Erlenmeyer flask containing 50 mL of ISP-4 and RA medium followed by culturing for 7 days at 200 rpm and 28 °C. Fermentations were extracted with butanone (1 × 50 mL) to get crude extracts; concentrates were then redissolved in 200 μL of MeOH and centrifuged at 12,000 g for 10 min to achieve clarified supernatant. High-performance liquid chromatography (HPLC) analyses were performed with an Agilent 1260 HPLC (Agilent Technologies, Santa Clara, CA, USA) using a linear gradient of 0% → 100% solvent B (solvent B: 0.1% HOAc-85% CH_3_CN in H_2_O; solvent A: 0.1% HOAc-15% CH_3_CN in H_2_O) over 30 min at a flow rate of 1 mL/min and monitored at 254 nm.

### 4.8. Aborycin Isolation

The filtrate of the culture broth (15 L) was extracted with butanone. Solvent removal in vacuo gave an oily yellowish residue (10.6 g), which was subjected to column chromatography over silica-gel (SiO_2_; 200-300 mesh; Qingdao Marine Chemical Ltd, Qingdao, China) eluted with CHCl_3_/MeOH mixtures of increasing polarity (100:0, 98:2, 96:4, 95:5, 90:10, 80:20, and 50:50, *v*/*v*). Column chromatography in this fashion afforded seven fractions (F1,1.2 g; F2, 2.6 g; F3, 4.9 g; F3, 1.2 g F4, 1.3 g; F5, 1.3 g; F6, 2.3 g and F7, 0.8 g); fraction F3 was further fractionated over silica gel using (96:4 CHCl_3_/MeOH) to give the major aborycin containing fraction (356 mg). Analytically pure aborycin was ultimately generated from this 356 mg sample by semi-preparative HPLC with an ODS column using an elution system consisting of solvent A (CH_3_CN) and solvent B (H_2_O), eluting over the course of 25 min (2.5 mL/min) to yield pure aborycin (30.5 mg).

### 4.9. Antibacterial Activities Assay

MIC values for aborycin were assessed using a 96-well plate format with MH broth. Methicillin-resistant *S. aureus* (MRSA), *E. faecalis*, *B. thuringiensis*, *Vibrio alginolyticus*, *E. gallinarum*, *Clostridium perfringens*, *Micrococcus luteus*, *E. coli*, *A. baumannii* and *K. pneumoniae* were selected for antibacterial activities assay. As previously described [11], briefly, the aborycin was first dissolved in DMSO at a concentration of 3.2 mg/mL; a 2 μL sample was serially diluted in 98 μL of MH broth. Then sequential 2-fold serial dilutions of the mix were serially diluted, where 50 μL MH broth and 50 μL cell cultures were added to wells. After incubation at 37 °C for 16~18 h, the MIC values of aborycin were determined in duplicate. Ampicillin, kanamycin, ciprofloxacin and polymyxin B were used as positive controls, respectively. 

## Figures and Tables

**Figure 1 marinedrugs-17-00127-f001:**
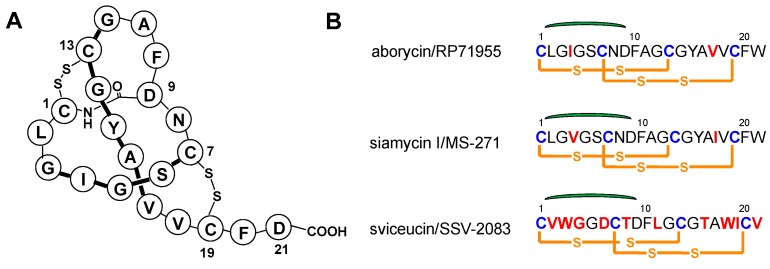
(**A**) Structure of aborycin. (**B**) Comparative sequences of primary structures of aborycin, siamycin and sviceucin.

**Figure 2 marinedrugs-17-00127-f002:**
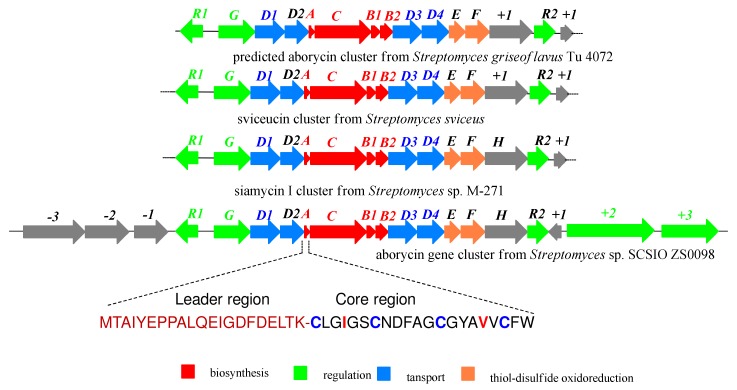
General BGC layout for type I lasso peptides (predicted aborycin cluster from *Streptomyces griseoflavus* Tü 4072, sviceucin cluster from *Streptomyces sviceus* and M-271 cluster from *Streptomyces* sp. are reproduced with permission from [5]) John Wiley & Sons, Inc., 2019, and aborycin gene cluster from *Streptomyces* sp. SCSIO ZS0098 and sequence of core region.

**Figure 3 marinedrugs-17-00127-f003:**
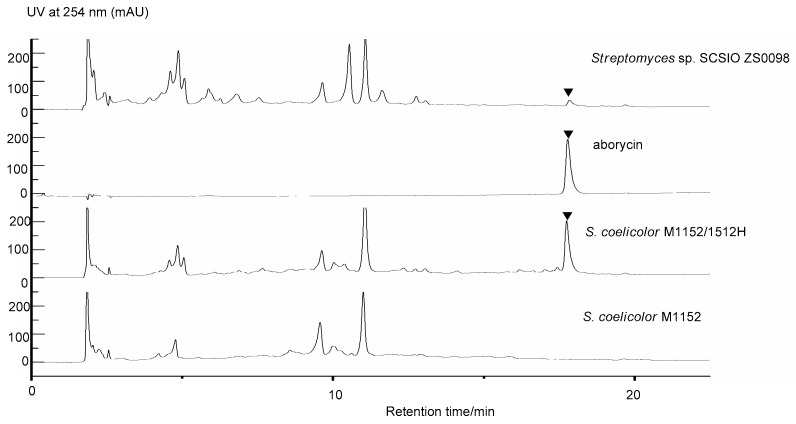
Comparative HPLC analysis of the aborycin and secondary metabolites in the culture extracts of wild-type *Streptomyces* sp. SCSIO ZS0098, *Streptomyces coelicolor* M1152/1512H and *S. coelicolor* M1152. ▼aborycin.

**Table 1 marinedrugs-17-00127-t001:** Proposed functions of ORF products encoded within the *abo* cluster.

ORF	Size ^a^	Closest Protein Similarity	ID/SI (%)	Accession Numbers
*orf(-3)*	688	AAA-like domain protein putative ATPase	34/50	ALJ40332
*orf(-2)*	466	nuclease	26/44	WP-045436186
*orf(-1)*	356	D12 class N6 adenine-specific DNA methyltransferase	44/55	ACU76144
*aboR1*	233	helix-turn-helix transcriptional regulator	99/99	WP-053077004
*aboG*	369	integral membrane sensor signal transduction histidine kinase	72/81	KPI21044
*aboD1*	313	multidrug ABC transporter ATP-binding protein	74/82	WP-059005878
*aboD2*	250	ABC transporter integral membrane protein	44/61	EDY58504
*aboA*	42	aborycin family tricyclic lasso peptide	98/100	KPI21041
*aboC*	603	lasso peptide isopeptide bond-forming cyclase	99/99	WP_114873958.1
*aboB1*	87	lasso peptide biosynthesis PqqD family chaperone	100/100	WP-037640857.1
*aboB2*	141	lasso peptide biosynthesis B2 protein	99/99	WP-048459500.1
*aboD3*	321	ABC transporter ATP-binding protein	75/85	KUN45102
*aboD4*	289	ABC transporter	98/98	WP-048459502
*aboE*	157	DoxX family protein	56/69	KWV32827
*aboF*	241	disulfide bond formation protein DsbA	98/99	WP-102640382.1
*aboH*	448	poly-gamma-glutamate biosynthesis protein	46/60	WP-038038039
*aboR2*	225	DNA-binding response regulator	99/99	WP-048459506
*orf(+1)*	116	Membrane protein involved in colicin uptake	51/59	CEL20868
*orf(+2)*	986	SARP family transcriptional regulator	92/93	WP-052183254
*orf(+3)*	632	transcriptional regulator	99/99	WP-048459508

^a^ Size in units of amino acids (aa); ID/SI: identity/similarity; *abo*: the BGC of aborycin from *Streptomyces* sp. SCSIO ZS0098.

**Table 2 marinedrugs-17-00127-t002:** Selected MICs for aborycin against assorted pathogenic bacteria (in μg/mL).

Pathogens	MIC (μg/mL)				
	Aborycin	Ampicillin ^a^	Kanamycin ^a^	Ciprofloxacin ^a^	Polymyxin B ^a^
*Staphylococcus aureus ATCC 29213*	8.0	16.0	2.0	^b^ NT	NT
*MRSA/methicillin-resistant Staphylococcus aureus*	8.0	>128	>128	NT	NT
MRSE*/methicillin-resistant Staphylococcus epidermidis*	128	>128	>128	NT	NT
*Staphylococcus aureus* Sau 29213	8.0	32.0	4.0	NT	NT
*Staphylococcus aureus* Sau 1862	16	>128	128.0	NT	NT
*Staphylococcus aureus* Sau 669	32	>128	1.0	NT	NT
*Staphylococcus aureus* Sau 991	16	>128	32.0	NT	NT
*Staphylococcus aureus* 16339	16	>128	2.0	NT	NT
*Staphylococcus aureus* 6917	16	>128	>128	NT	NT
*Staphylococcus aureus* 16162	64	>128	>128	NT	NT
*Staphylococcus aureus* 718306	16.0	>128	2.0	NT	NT
*Staphylococcus aureus* 745524	16.0	>128	>128	NT	NT
*Staphylococcus aureus* MRSA GDE4P037P	>128	>128	>128	NT	NT
*Staphylococcus aureus* (cfr) GDQ6P012P	64	>128	4.0	NT	NT
*Staphylococcus cohnii* DKG4	>128	>128	>128	NT	NT
*Staphylococcus simulans* AKA1	>128	>128	>128	NT	NT
*Vibrio alginolyticus* XSBZ14	>128	>128	4.0	NT	NT
*Enterococcus faecalis* ATCC 29212	8.0	>128	>128	NT	NT
*Enterococcus gallinarum* 5F52C	0.5	8.0	8.0	NT	NT
*Clostridium perfringens* FSKP20	>128	>128	>128	NT	NT
*Micrococcus luteus*	>128	2.0	0.5	NT	NT
*Bacillus thuringiensis*	2.0	>128	64.0	NT	NT
*Escherichia coli* 16369	>128	NT	NT	32.0	4.0
*Escherichia coli* 16447	>128	NT	NT	0.25	4.0
*Escherichia coli* 737720	>128	NT	NT	8.0	4.0
*Escherichia coli* 16351	>128	NT	NT	64.0	4.0
*Escherichia coli* ATCC 13124	>128	NT	NT	32.0	>128
*Escherichia coli* (E11)	>128	NT	NT	8.0	2.0
*Acinetobacter baumannii* ATCC 19606	128.0	NT	NT	8.0	4.0
*Acinetobacter baumannii* 15122	>128	NT	NT	32.0	4.0
*Acinetobacter baumannii* 15407	>128	NT	NT	32.0	4.0
*Acinetobacter baumannii* 15199	>128	NT	NT	32.0	4.0
*Acinetobacter baumannii* 14892	>128	NT	NT	16.0	4.0
*Klebsiella pneumoniae* ATCC 13883	>128	NT	NT	0.25	4.0
*Klebsiella pneumoniae* 15580	>128	NT	NT	4.0	2.0

^a^ ampicillin, ^a^ kanamycin, ^a^ ciprofloxacin and ^a^ polymyxin B were assayed as a positive control. ^b^ NT Not tested.

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
