# Peer review of "Identification of the Anti-Infective Aborycin Biosynthetic Gene Cluster from Deep-Sea-Derived Streptomyces sp. SCSIO ZS0098 Enables Production in a Heterologous Host"

_marinedrugs, 2019, doi:10.3390/md17020127_

Round 1

Reviewer 1 Report

An excellent paper with more than adequate details given. 

Two very minor points.

Line 89, remove second "n" from "containning"

Lines 172-3, the paper refers to a deep-sea sediment. Please give the depths at which the sediment was collected

Author Response

Response to Reviewer #1:
Comments and Suggestions for Authors

An excellent paper with more than adequate details given. 

Response: Thanks for your comments!

Two very minor points.

Line 89, remove second "n" from "containning"

Response: Thanks for your suggestion! We have corrected accordingly.

Lines 172-3, the paper refers to a deep-sea sediment. Please give the depths at which the sediment was collected

Response: We have added the corresponding data in the bacterial strains and plasmids section.

Reviewer 2 Report

A well presented and reasonably interesting study that will be of sufficient interest to the wider natural products community to merit publication. The paper is well written and presented and the work appears sound.

Author Response

Response to Reviewer #2:

Comments and Suggestions for Authors

A well presented and reasonably interesting study that will be of sufficient interest to the wider natural products community to merit publication. The paper is well written and presented and the work appears sound.

Response: Thanks for your comments!

Reviewer 3 Report

The manuscript represents a preliminary study to prepare some antiinfective compounds useful to work where other antibiotics are not more effective (e.g. kanamycin resistance)!!

The authors should explain something regarding Aborycin mechanisms of action and improve the English along the entire manuscript.

Author Response

Response to Reviewer #3:

Comments and Suggestions for Authors

The manuscript represents a preliminary study to prepare some antiinfective compounds useful to work where other antibiotics are not more effective (e.g. kanamycin resistance)!!

Response: Thanks for your comments!

The authors should explain something regarding Aborycin mechanisms of action and improve the English along the entire manuscript.

Response: We appreciate this point by reviewer 3! As suggested, we added in the main text sentence Aborycin shares some sequence identity with the HIV-1 protease gp41 which belongs to the aspartic proteinase family, thus inhibiting HIV-1 replication [1,3]. referred to the references ([1] Helynck, G., Dubertret, C., Mayaux, J. F., Leboul, J. Isolation of RP 71955, a new anti-HIV-1 peptide secondary metabolite. J. Antibiot. 1993 11, 1756-1757; [3] Frechet, D., Guitton, J. D., Herman, F., Faucher, D., Helynck, G., Monegier du Sorbier, B., Vuilhorgne, M. Solution structure of RP 71955, a new 21 amino acid tricyclic peptide active against HIV-1 virus. Biochemistry 1994, 1 42-50). 

Reviewer 4 Report

This study is presented as the first report of the cloning of a gene cluster responsible for the synthesis of the lasso peptide aborycin in Streptomyces sp.

Following the identification of a putative biosynthetic cluster by in silico analyses, the authors carried out its heterologous cloning (from a cosmid library) and its expression in Streptomyces coelicolor. Antibacterial activity of aborycin was demonstrated against a large pannel of pathogenic bacterial species.

The manuscript is nicely written, the methodology is correctly used, and the conclusions are supported by the data.

Line 60: abroycin should read aborycin

Author Response

Response to Reviewer #4:

Comments and Suggestions for Authors

This study is presented as the first report of the cloning of a gene cluster responsible for the synthesis of the lasso peptide aborycin in Streptomyces sp. Following the identification of a putative biosynthetic cluster by in silico analyses, the authors carried out its heterologous cloning (from a cosmid library) and its expression in Streptomyces coelicolor. Antibacterial activity of aborycin was demonstrated against a large pannel of pathogenic bacterial species. The manuscript is nicely written, the methodology is correctly used, and the conclusions are supported by the data.

Response: We appreciate these comments!

Line 60: abroycin should read aborycin

Response: We have corrected accordingly.

Reviewer 5 Report

Minor points:

line 21: remove dot after Bacillus

line 56-59: sentence structure needs improvement; rephrase sentences

line 64: correct abbreviation: S. sp. SCSIO 

line 95-96: I think you can remove the comma after that 

line 261: double space

line 262: correct: .,

introduction: a recent review about the topic lassopeptides, biosynthesis, structures is missing 

Author Response

Response to Reviewer #5:

Comments and Suggestions for Authors

Minor points:

line 21: remove dot after Bacillus

line 56-59: sentence structure needs improvement; rephrase sentences

line 64: correct abbreviation: S. sp. SCSIO 

line 95-96: I think you can remove the comma after that 

line 261: double space

line 262: correct: .,

Response: We have corrected above all accordingly.

introduction: a recent review about the topic lassopeptides, biosynthesis, structures is missing

Response: Thanks for your suggestion! We have added as reference #4 (Maksimov, Mikhail O., PAN, Si Jia, Link, A. James. Lasso peptides: structure, function, biosynthesis, and engineering. Nat. Prod. Rep. 2012, 9, 996-1006.) in our revised manuscript.

Reviewer 6 Report

This original research article submitted by Mingwei Shao et al., entitled “Identification of the Anti-infective Aborycin Biosynthetic Gene Cluster from Deep Sea-derived Streptomyces sp. SCSIO ZS0098 Enables Production in a Heterologous Host” deals about the discovery of aborycin, a natural bioactive lasso peptide in a new Streptomyces strain and its expression in an heterologous host. The manuscript is clear and very well written; however, I consider that at the present time the manuscript requires some minor corrections, additional information and modifications before acceptance.

Editing:

Line 33: please detail what do you mean by “structure”

Line 60: Please correct “abroycin”

Line 83: Figure 2: please consider using another colour than yellow for your figure (orange, pink,…). Also include references in the legend (Adapted from….”REFERENCE”) for other BGC layouts (Streptomyces griseoflavus, Strept. sviceus and M-271)

Line 170: Please add reference for Zhi Feng and co-workers.

Line 171: Please add reference for Yanyan Li and co-workers.

Line 196: Please add information (volume, inoculation, static/rpm) about conditions for sporulation.

Line 223: Please replace “cosmids” by “cosmid” and correct “were” consequently.

Line 226: Please replace “cosmids” by “cosmid”.

Lines 232, 233 please use “day/s” and not “d”.

Line 240: Please change “The Isolation of the Aborycin” by “Aborycin isolation”.

Line 252: Please list all strains used for the antibacterial screening.

Figure S1: Please correct “aboycin”

Comments:

Line 52 – 56:  Results of preliminary screening must be included in the manuscript (table in supplementary data). The results must be rewritten and please do not use the expression “for example” (line 54), which is absolutely not scientific, please list all results (positive and negative).

Author Response

Response to Reviewer #6:

Comments and Suggestions for Authors

This original research article submitted by Mingwei Shao et al., entitled “Identification of the Anti-infective Aborycin Biosynthetic Gene Cluster from Deep Sea-derived Streptomyces sp. SCSIO ZS0098 Enables Production in a Heterologous Host” deals about the discovery of aborycin, a natural bioactive lasso peptide in a new Streptomyces strain and its expression in an heterologous host. The manuscript is clear and very well written; however, I consider that at the present time the manuscript requires some minor corrections, additional information and modifications before acceptance.

Response: Thanks for your comments! 

Editing:

Line 33: please detail what do you mean by “structure”

Response: Thanks for your suggestion! We have replace the sentence now to read Aborycin shares some sequence identity with the HIV-1 protease gp41 which belongs to the aspartic proteinase family, thus inhibiting HIV-1 replication [1,3].

Line 60: Please correct “abroycin”

Response: We have corrected accordingly.

Line 83: Figure 2: please consider using another colour than yellow for your figure (orange, pink,…). Also include references in the legend (Adapted from….”REFERENCE”) for other BGC layouts (Streptomyces griseoflavusStrept. sviceus and M-271)

Response: Thanks for your suggestion! We have changed the yellow to orange and  include references in the legend (predicted aborycin cluster from Streptomyces griseoflavus Tü 4072, sviceucin cluster from S. sviceus and M-271 cluster from Streptomyces sp. are adapted from reference [5]).

Line 170: Please add reference for Zhi Feng and co-workers.

Line 171: Please add reference for Yanyan Li and co-workers.

Line 196: Please add information (volume, inoculation, static/rpm) about conditions for sporulation.

Line 223: Please replace “cosmids” by “cosmid” and correct “were” consequently.

Line 226: Please replace “cosmids” by “cosmid”.

Lines 232, 233 please use “day/s” and not “d”.

Line 240: Please change “The Isolation of the Aborycin” by “Aborycin isolation”.

Response: We have corrected above all accordingly.

Line 252: Please list all strains used for the antibacterial screening.

Response: Thanks for your suggestion! We have listed all strains used for the antibacterial screening in our revised manuscript.

Figure S1: Please correct “aboycin”

Response: We have corrected accordingly.

Comments:

Line 52 – 56:  Results of preliminary screening must be included in the manuscript (table in supplementary data). The results must be rewritten and please do not use the expression “for example” (line 54), which is absolutely not scientific, please list all results (positive and negative).

Response: Thanks for your suggestion! We have listed all strains used for the antibacterial screening in our revised manuscript.